# Hemophagocytic Lymphohistiocytosis Associated with Synergistic Defects of *AP3B1* and *ATM* Genes: A Case Report and Literature Review

**DOI:** 10.3390/jcm12010095

**Published:** 2022-12-22

**Authors:** Guangjiao Yin, Yasu Lu, Huaqin Pan, Bin Deng, Sanyun Wu, Zhiyong Peng, Xujun Ye

**Affiliations:** 1Department of Geriatrics, Zhongnan Hospital of Wuhan University, Wuhan 430071, China; 2Department of Critical Care Medicine, Zhongnan Hospital of Wuhan University, Wuhan 430071, China; 3Clinical Research Center of Hubei Critical Care Medicine, Wuhan 430071, China; 4Department of Hematology, Zhongnan Hospital of Wuhan University, Wuhan 430071, China

**Keywords:** hemophagocytic lymphohistiocytosis, sepsis, multiorgan dysfunction, shock, gene mutation, intensive care unit

## Abstract

Hemophagocytic lymphohistiocytosis (HLH) is an overwhelming immune system activation that manifests as hyperinflammation and life-threatening multiple organ failure. However, the clinical manifestations of the systemic inflammatory response in sepsis and fulminant cytokine storm caused by HLH macrophage activation are very similar and difficult to distinguish. HLH triggered by two novel gene defects manifesting with multiorgan dysfunction syndrome (MODS) and distributive shock has not been reported. A 14-year-old male patient was hospitalized with a high fever, his condition deteriorated rapidly, accompanied by cytopenia, shock, and MODS, and he was subsequently transferred to our intensive care unit (ICU) for symptomatic and organ-supportive treatments. Laboratory indicators of cytopenia, hypofibrinogenemia, hypertriglyceridemia, hyperferritinemia, high soluble CD25, low natural killer (NK) cell cytotoxicity, and hemophagocytosis in the bone marrow confirmed the diagnosis of HLH. Molecular genetic analysis revealed that two novel heterozygous gene mutations in *AP3B1* (c.3197 C > T) and *ATM* (c.8077 G > T) might have accounted for the onset. After treatment, the patient’s condition successfully improved. This case report demonstrates the timely determination of underlying triggers and critical care supports (supportive and etiological treatment) of HLH related to the improved outcome.

## 1. Introduction

Hemophagocytic lymphohistiocytosis (HLH) is a lethal hyperinflammatory syndrome caused by primary or secondary factors. It is also referred to as hemophagocytic syndrome (HPS) and is characterized by persistent high fever, hepatosplenomegaly, hemocytopenia, elevated transaminase and ferritin levels, and coagulation disorders [1,2,3]. Diagnosis of HLH is challenging precisely because its clinical manifestations may be indistinguishable from hyperinflammatory disorders, such as sepsis. The estimated incidence of HLH ranges from 1 to 225 cases per 300,000 live births [4]. In comparison, the overall frequency of HLH in critical illness is unknown but rare [5]. The mortality rate for refractory HLH is as high as 70% [6].

Multiple causes can trigger HLH, for example, gene mutation. Depending on different causes, HLH can be divided into primary and secondary HLH. Primary HLH is an autosomal or sex chromosomal recessive disorder closely associated with impaired genes related to lymphocytotoxicity or inflammasome activation [1]. Most children with primary HLH have an identifiable genetic defect that manifests as a Mendelian pure or compound heterozygous lesion [7]. Familial HLH (FHL) is the most common type of primary HLH, with perforin-mediated lymphocyte cytotoxicity gene defects in *PRF1*, *UNC13D*, *STX11*, and STXBP2. Notably, several primary immunodeficiency syndromes are predisposed to HLH. Specifically, Griscelli syndrome type 2 (GS-2), Chediak–Higashi syndrome 1 (GHS-1), and Hermansky–Pudlak syndrome type 2 (HPS-II) are associated with several granule/pigment abnormality genes, *RAB27A*, *LYST*, and *AP3B1*, respectively [1]. Secondary HLH is caused by a variety of triggers, including infections, malignancies, macrophage activation syndrome (MAS), autoimmune diseases, pregnancy, drugs, hematopoietic stem cell transplantation, and rare metabolic disorders [8,9]. Primary HLH is more common in children, with a median age of 1.8 years, and secondary HLH is more common in adults, with a mean age of approximately 50 years [7]. However, the incidence of primary HLH in adolescents and adults is increasing [10].

Thirty-two HLH cases with biallelic mutations in *AP3B1* have been reported in the literature, with one patient developing lethal HLH [11], while the *ATM* gene was only reported in a 3-year-old male infant [12]. Herein, we report a case of a 14-year-old male diagnosed with fulminant HLH that might be associated with *AP3B1* and *ATM* genes’ mutation. Most importantly, our case provides clinical experience for the successful treatment of critically ill HLH patients with a genetic mutation background.

## 2. Case Presentation

A 14-year-old male patient presented to our hospital with a persistent fever of up to 40 °C, accompanied by a sore throat and watery diarrhea. Oral herpes developed on day 3 of the illness. The patient was initially treated empirically with imipenem–cilastatin sodium at a local clinic. On the third day of the illness, his condition deteriorated rapidly, and he was transferred to the intensive care unit (ICU).

The patient’s consciousness was clear, body temperature 38.1 °C, heart rate 101 beats per minute, respiratory rate 26 breaths per minute, blood pressure 104/58 mmHg maintained by norepinephrine and oxygen saturation in high-flow nasal oxygenation (HFNO) 94% for circulatory and respiratory supports. HFNO flow was 20 L/min, and FiO2 was 50%. Chest examination revealed wet rales at the base of both lungs. The patient had been susceptible to upper respiratory tract diseases such as tonsillitis and influenza since childhood and had no family history of genetic disease. The clinical outcome data were summarized as follows:White blood cell (WBC) and platelet (PLT) counts decreased dramatically. Hemoglobin (HGB) concentration was also reduced.Multiorgan dysfunction (including acute liver failure, acute kidney injury, and myocardial injury). Levels of alanine aminotransferase (ALT), aspartate aminotransferase (AST), blood urea nitrogen (BUN), creatinine (CREA), myocardial enzyme, and brain natriuretic peptide (BNP) were noticeably elevated.Respiratory failure with PaO2/FiO2 of 220 mmHg.Prolonged prothrombin time, markedly elevated D-dimer, and reduced fibrinogen levels.High inflammatory response dramatically increased procalcitonin (PCT), interleukin 6 (IL-6), C-reactive protein (CRP), and ferritin.High triglyceride (TG) (Table 1).The chest computed tomography (CT) scan showed inflammatory exudate and oedema in both lungs, bilateral pleural effusion, enlarged liver and spleen, and brain cell oedema (Figure 1).Within 24 h of admission to the ICU, the scores of acute physiology and chronic health evaluation-II (APACH II), sequential organ failure assessment (SOFA), and nutrition risk screening-2002 (NRS-2002) were 24, 15, and 5, respectively.

The patient received endotracheal intubation for invasive mechanical ventilation (IMV), norepinephrine and intravenous rehydration for blood circulation, plasma exchange (PE) for liver function, continuous renal replacement therapy (CRRT) for renal function support, and multiple blood products (fresh frozen plasma, platelets, and coagulation factors) for improving coagulation.

Sepsis caused by the infection in this patient was first considered. Based on the clinical experience, prophylactic treatments were initiated with antibiotics such as linezolid and carbapenem for anti-bacteria, ribavirin for antiviruses, voriconazole for anti-fungi, and sulfamethoxazole for anti-pneumocystis carinii infection. In addition, the cause of the infection was sought before broad-spectrum antibiotic treatment. Relevant laboratory tests were completed, cultures of blood, stool, urine, and bronchoalveolar lavage fluid (BALF) were performed, and pathogenic metagenomic testing of blood and BALF was performed to rapidly screen for potential pathogens.

In our case, the patient presented with a trilineage reduction in peripheral blood cells, persistent high fever, and splenomegaly. Laboratory tests revealed a sharp increase in ferritin, with a value of over 33,511 ng/mL. This prompted the consideration of hematologic diseases, followed by a bone marrow examination. Hemophagocytic cells were seen in the bone marrow (Figure 2). Therefore, the diagnosis of HLH was highly suspected. Further laboratory tests, such as tests for ferritin, nature kill cell activity, and soluble CD25, were performed after consultation with a hematologist. These laboratory results met eight of the diagnostic criteria for HLH (Table 2). Finally, HLH was diagnosed. During hospitalization, the patient developed central nervous system involvement with the clinical manifestation of epilepsy.

The diagnosis of HLH was established, and further causes were identified in the following aspects. Early recognition of the potential causes was critical to the treatment. Firstly, HLH was suspected to be caused by infection, but more than three times of cultures in blood, urine, stool, bone marrow, and BALF did not reveal any potential pathogens. The (1,3)-β-D-glucan test and galactomannan (GM) test in whole blood and BALF did not support a fungal infection. Metagenomic testing of blood and BALF for pathogens (based on the Illumina high-throughput sequencing platform with 18,562 microorganisms in the database) showed negative results for bacteria, fungi, viruses (DNA viruses and RNA viruses), parasites, mycobacterium tuberculosis, mycobacterium non-tuberculosis, pneumocystis carinii, mycoplasma, and chlamydia. Blood laboratory tests were negative for influenza A, influenza B, Epstein–Barr virus (EBV), cytomegalovirus (CMV), respiratory syncytial virus (HSV), coxsackievirus, and herpes simplex virus, and positive only for the influenza B virus IgM antibody. Secondly, the patient’s autoimmune antibodies, including the antinuclear antibody (ANA), extractable nuclear antigen (ENA), antineutrophil cytoplasmic antibody (ANCA), and anti-cardiolipin antibody, were normal. HLH-related rheumatoid immune diseases were excluded. Based on the laboratory results of the bone marrow, leukemia, lymphoma, and multiple myeloma were not considered. In terms of malignancy, lymph node changes caused by HLH were presented according to the pathological morphology and immunohistochemical results of the lymph nodes (Figure 3a). The tumor markers of this patient were all negative. Positron emission computed tomography (PET/CT) suggested bilateral axillary and retroperitoneal multiple small lymph nodes with nonspecific inflammatory hyperplasia and hepatomegaly. No obvious hypermetabolic metastases were observed in the body (Figure 3b). The patient was not taking any medication prior to the onset, thus ruling out drug-induced secondary HLH.

Secondary causes were ruled out, and primary HLH was highly suspected; most likely a genetic mutation. On the 9th day of admission, the phagocytic syndrome expanded mutation gene screening test in bone marrow reported *AP3B1*: NM_003664.4: exon27: c.3197C > T: p. Ser1066Phe, with a variation ratio of 49.21%. The primary HLH was initially considered based on the above adjuvant findings.

On 19 November 2021, chemotherapy with etoposide (VP-16) (0.15 g twice weekly) was administered after a multidisciplinary discussion. After the second chemotherapy on 23 November, the patient’s symptoms continued to alleviate, and he was then transferred to the general hematology ward for cyclical chemotherapy. The specific timeline of the patient’s treatment in the ICU is shown below (Figure 4). After three months of follow-up, the patient’s HLH-related clinical symptoms, imaging manifestations, and laboratory indicators were normalized.

## 3. Materials and Methods

After obtaining informed consent from the child and his parents, whole exome sequencing (WES) was performed on the DNA extracted from peripheral blood using next-generation sequencing (NGS) technology (testing equipment: Hiseq X Ten, Novaseq). The patient was confirmed to have both *AP3B1* and *ATM* gene mutations, with his father having an *AP3B1* gene mutation and his mother having an *ATM* gene mutation. The synergistic function of heterozygous mutations in *AP3B1* and *ATM* may lead to HLH.

A novel heterozygous missense mutation was detected in the *AP3B1* gene: NM_003664.4: exon27: c.3197C > T (p. Ser1066Phe), which was analyzed by genealogical verification and was derived from his father. A heterozygous missense mutation was detected in exon 55 of the *ATM* gene: NM_000051.3: exon55: c.8077G > T (p. Ala2693Ser), which was analyzed by genealogical verification and was derived from his mother. Newly identified mutations were confirmed in the proband and his parents by using Sanger sequencing (Figure 5). The details of the WES and Sanger sequencing data are given in the Appendix A. In Figure 5, the *ATM* gene needs to be clarified as the heterozygous mutation either in his mother or the patient. We performed a TA clone. Sequencing results showed that the father was all wild-type, and the mother and the patient had both wild-type and mutant (Appendix A). The primer and sequencing results are shown in Appendix A. Synergistic mutations in both may lead to impaired protein function, which in turn contributes to the development of HLH.

## 4. Discussion

HLH is difficult to distinguish from sepsis, MODS, and other cytokine storm syndromes, but the severity of the disease is often beyond the expectations of critical illness [13]. It is caused by extreme activation of cytotoxic T lymphocytes (CTLs), natural killer (NK) cells, and macrophages [7,14]. Immunodeficient susceptible hosts undergo significant immune activation, resulting in cytokine-mediated tissue injury and multiorgan dysfunction [15]. Given the nonspecific clinical findings of HLH, this makes the diagnosis of patients with a genetic background of susceptibility challenging.

If the HLH-2004 diagnostic criteria are met in five of eight items, then HLH can be diagnosed [16]. However, some of these indicators, such as a bone marrow biopsy, NK cell activity, and soluble CD25, are not readily available. Hyperferritinemia is an important diagnostic criterion for HLH and has been regarded as an efficient and convenient indicator for suspected HLH cases. The HLH-2004 guidelines include ferritin over 500 μg/L as one of the diagnostic criteria, with a sensitivity of 84% [16]. Ferritin is an acute phase reactant that is more common in inflammation and infection. Extensive macrophages activate and release intracellular ferritin, which may be responsible for the significant increase in ferritin levels [17]. Elevated ferritin levels are seen in various conditions, such as sepsis, stroke, acute respiratory distress syndrome, renal failure, malignancy, liver disease, alcohol excess, and metabolic syndrome [18,19]. Notably, unlike patients with sepsis, septic shock, and other conditions, patients with HLH have the highest levels of ferritin. In a retrospective study, the cutoff value of ferritin level was 9083 μg/L (sensitivity 92.5%, specificity 91.9%) to distinguish hyperferritinemia caused by HLH and sepsis or other diseases in a mixed ICU cohort [20]. The cutoff value for ferritin diagnosis varies for patients with HLH of different age groups. A ferritin over 10,000 μg/L is 90% sensitive and 96% specific for the diagnosis of HLH in children [17]. In adults, drastically elevated ferritin levels are more meaningful for diagnosing HLH. A retrospective observational study indicated that ferritin levels above 10,000 μg/L and 100,000 μg/L predicted approximately 14% and 61% of adult patients with HLH, respectively [21]. In our case, the patient presented with fever, cytopenia, splenomegaly, distributed shock, MODS, and other nonspecific clinical manifestations. Laboratory results showed a dramatic increase in ferritin (>33,511 ng/mL). Ferritin has important predictive value in the diagnosis of HLH. HLH can be diagnosed early when elevated serum ferritin is associated with fever, cytopenia, and abnormal changes in the liver and spleen. Meanwhile, in view of the patient’s unexplained and rapidly progressing inflammatory responses, HLH was first considered and further confirmed by follow-up laboratory findings. Therefore, HLH should be highly suspected when uncontrolled inflammation occurs in ICU. At the same time, we actively searched for the triggers and found that the patient had *AP3B1* and *ATM* defects, which provided guidance for treatment. The patient’s symptoms were relieved after chemotherapy (etoposide) through intensive supportive treatment.

In our case, mutations in the *AP3B1* and *ATM* genes were highly suspected as potential triggers. The *AP3B1* gene is located on the long arm of chromosome 5 (5q14.1). The adaptor protein 3 (AP-3) complex, encoded by the *AP3B1* gene, plays an important role in transporting the cargo proteins from the Golgi and tubular endosomal compartment to endosome–lysosome-related organelles [22,23,24]. *AP3B1* mutation causes loss of function in the aforementioned vesicular trafficking, including melanosomes and platelet dense granules [24,25]. When *AP3B1* homozygous mutation leads to the onset of HPS-II [26], it manifests as decreased pigmentation (albinism) with visual impairment, blood platelet dysfunction with prolonged bleeding, and pulmonary fibrosis [27]. Although *AP3B1* gene deficiency impairs cytotoxic T lymphocyte or NK cell degranulation and/or cytotoxicity, the clinical phenotypic severity is variable according to impairment level [11]. To some extent, there is an increased risk of HLH [11,28]. Fabiola Dell’Acqua et al., describe the *AP3B1* gene and its structure in 27 exons and summarize all biallelic mutations published so far [11]. In this case, an AP3B1 de novo mutation did not lead to HPS-II. Genetic testing analysis was performed, and a concurrent combination of mutations in the *ATM* gene that induced HLH was found. Consistent with this, mutations in the *AP3B1* gene are less common in patients with HLH; when combined with other unknown gene mutations or genes encoding that are involved in the lytic granule exocytosis pathway in NK/CTLs, it leads to the development of primary HLH [29]. *AP3B1* synergistic mutations in *Rab27A* and *UNC13D* genes have been demonstrated to cause severe primary/secondary HLH [28,30].

Ataxia-telangiectasia (A-T) is a rare autosomal recessive disorder caused by mutations in the A-T mutated (*ATM*), often resulting in immunodeficiency and susceptibility to hematological malignancies [31,32,33]. *ATM* is localized at 11q22-23, and the protein encoded by the gene belongs to the phosphoinositide 3/4-kinase-related kinases (PI3/PI4) family. It is an essential phosphorylated cell cycle checkpoint kinase that acts as a regulator of various downstream proteins [34]. Molecularly, the *ATM* gene acts as a checkpoint in the DNA damage response network, promoting cell cycle arrest and apoptosis [35,36]. Germ-line mutations in *ATM*, leading to genomic instability, defective DNA double-strand repair, and selective escape from apoptosis, can be seen as a DNA replication stress that promotes the development and progression of cancer [32,36,37]. Mehmet H et al., reported a 3-year-old male infant with A-T who developed HLH during follow-up [12]. Although there is no direct evidence that mutations in *AP3B1* and *ATM* cause HLH, both are associated with HLH. In our case, the synergistic effect of these two genetic mutations might have been responsible for the activation of HLH. The causality between the observed mutations and the genesis of HLH requires further functional studies to verify.

For HLH in critical illness, the trigger should be identified as early as possible in conjunction with intensive interventions such as broad-spectrum antibiotics, vasopressors, positive pressure ventilation, blood product replacement therapy, renal replacement therapy, and even extracorporeal life support. Prompt initiation of treatment directed at the underlying trigger is essential. Selecting the optimal treatment for HLH is vital and usually based on the guidelines of the HLH-1994 protocol [5,38,39]. Standard HLH initial treatment consists of an 8-week course of etoposide (150 mg/m^2^ twice weekly for two weeks, then weekly) plus dexamethasone (initial dose of 10 mg/m^2^, tapering over eight weeks) [4]. In our case, the patient’s symptoms significantly improved after receiving supportive care and two etoposide treatments. Given his refusal to receive Allo-HSCT, there is still a risk of recurrence at any time. Allo-HSCT remains the only long-term curative therapy [40]. HLH-associated biallelic mutations determine whether patients have clear indications for transplantation [16]. A study shows that the prognosis of allogeneic hematopoietic stem cell transplantation before the onset of HLH symptoms is significantly better than that of those who have developed HLH [41]. The HLH Steering Committee of the Histiocyte Society pointed out that for asymptomatic HLH-related biallelic mutation carriers, if HLH has manifested in a family member in infancy, HSCT is strongly recommended as soon as possible [42]. In the future, for HLH with established biallelic mutations, allogeneic hematopoietic stem cell transplantation before symptom onset may lead to a better prognosis. However, large-scale prospective cohort studies are still needed.

## 5. Conclusions

The diagnosis of HLH is generally challenging due to its polymorphous and atypical clinical manifestations. Rapid diagnosis and aggressive management can improve the prognosis of critically ill patients. Furthermore, we report a case of HLH with a new mutation in the *AP3B1* gene: c.3197C > T (p. Ser1066Phe), combined with a mutation in the *ATM* gene: c.8077G > T (p. Ala2693Ser). Simultaneous mutations in both *AP3B1* and *ATM* genes may lead to impaired protein function, which in turn contributes to the development of fulminant HLH for the first time. Remarkably, the case was successfully treated in the ICU, and this case significantly expands the clinical understanding of fulminant HLH.

## Figures and Tables

**Figure 1 jcm-12-00095-f001:**
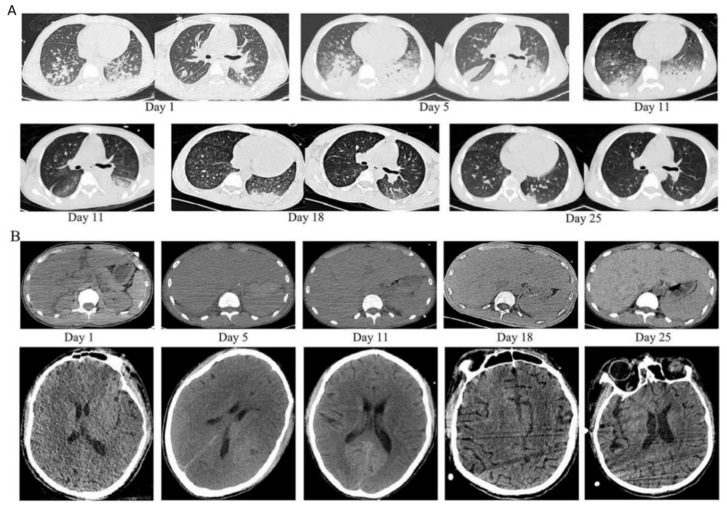
High-resolution computed tomography images during the disease course. (**A**) Upper panel: diffuse infection of both lungs with multiple patchy, nodular shadows, interstitial pulmonary oedema, and a small amount of pleural effusion bilaterally. (**B**) Low panel: enlarged liver and spleen size, brain cell oedema.

**Figure 2 jcm-12-00095-f002:**
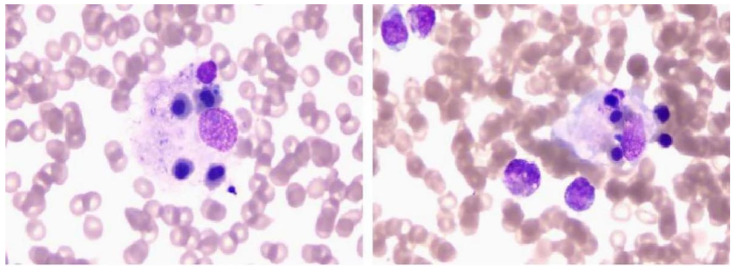
Bone marrow biopsy showed hemophilic cells that engulf leucocytes and nucleated red blood cells.

**Figure 3 jcm-12-00095-f003:**
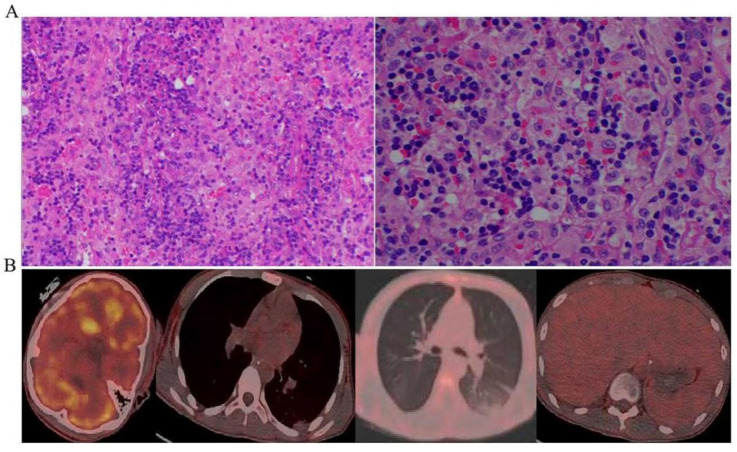
No signs of malignancy were seen in this patient. (**A**) Microscopically, the lymph node structure was destroyed, and no obvious lymphoid follicle structure was found. The interfollicular area and lymphatic sinus were enlarged. Significant histiocyte infiltration with phagocytosis of erythrocytes and lymphocytes by histiocytes. (**B**) PET-CT showed no signs of malignancy with abnormally increased metabolism.

**Figure 4 jcm-12-00095-f004:**
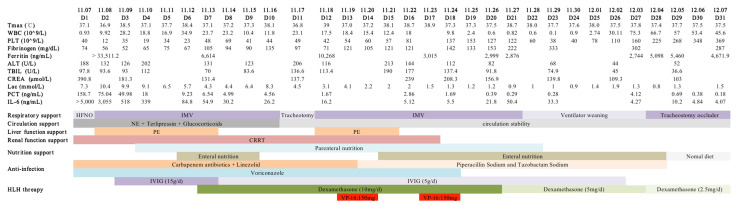
Timeline of the course of disease treatment based on the number of days in the ICU. From 7 November to 7 December 2021. Abbreviation: IMV, invasive mechanical ventilation; NE, norepinephrine; PE, plasma exchange; CRRT, renal replacement therapy; IVIG, intravenous immunoglobulin.

**Figure 5 jcm-12-00095-f005:**
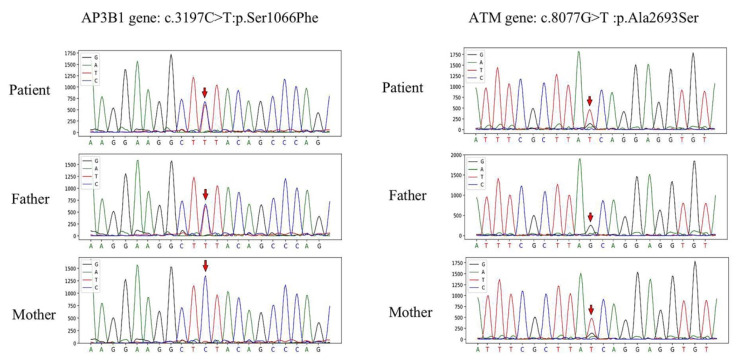
The mutation sites in this patient include the *AP3B1* variant c.3197C > T and the *ATM* gene c.8077G > T, both of which are newly identified variants, the former from the father and the latter from the mother.

**Table 1 jcm-12-00095-t001:** Clinical laboratory results.

Measure	Normal Range	11/07 Day 1	11/12 Day 5	11/17 Day 10	11/22 Day 15	11/27 Day 20	12/03 Day 25	12/08 Day 30
WBC (10^9^/L)	3.5–9.5	0.93	22.36	23.1	18	0.82	75.27	33
HGB (g/L)	130–175	101	91	81	82	81	75	82
PLT (10^9^/L)	125–350	40	23	49	81	122	160	409
NEUT (10^9^/L)	1.8–6.3	0.42	11.62	16.1	11.2	12.9	13.4	11.9
FIB (mg/dL)	238–498	74	112	96	121	222	302	287
PT (s)	9.4–12.5	31.2	11.9	11.6	11.2	12.9	13.4	11.9
APTT (s)	25.1–36.5	80.6	37.6	33.6	25.8	24.7	29	31.6
PTTA (%)	80–130	26	79	78	96	88	84	87
DD (ng/mL)	0–500	>35,000	>35,000	58,879	28,946	15,324	3641	2903
Ferritin (ng/mL)	21.8–274.7	>33,511	6614	10,268	3014	2876	2743	4671
TG (mmol/L)	<1.7	2.83						
ALT (U/L)	9–50	153	151	206	144	82	44	
AST (U/L)	15–40	661	994	658	199	57	38	
TBIL (μmol/L)	5–21	73.1	113.8	136.6	177	91.8	45	
BUN (mmol/L)	2.8–7.6	13.7			27.2	22.9	13	
CREA (μmol/L)	64–104	390.8			239	156.9		
PCT (ng/mL)	<0.05	158.68	9.23		2.86	0.29	4.12	0.18
IL-6 (ng/mL)	0–7	>5000	84.8		5.12	50.4	4.27	
BNP (pg/mL)	<100	177.1	196.7	23.9				
CK-MB (ng/mL)	0–6.6	21.1	9.3	1.4	1.8		4	
cTNI(pg/mL)	0–26.2	4740.5	368.3	414.6	139.1		5.2	

Abbreviation: WBC, white blood cell; HGB, hemoglobin; PLT, platelets; NEUT, neutrophil; FIB, fibrinogen; PT, prothrombin time; APTT, activated partial thromboplastin time; PTTA, prothrombin time activity; DD, D-Dimer; TG, triglyceride; ALT, alkaline phosphatase; AST, aspartate transaminase; TBIL, total bilirubin; BUN, blood urea nitrogen; CREA, creatinine; PCT, procalcitonin; IL-6, interleukin-6; BNP, brain natriuretic peptide; CK-MB, creatine kinase myocardial band; cTNI, cardiac troponin I.

**Table 2 jcm-12-00095-t002:** Eight of the diagnostic criteria for HLH.

At Least 5 of the Following 8 Findings	
Non-remitting fever ≥ 38.5 °C	Yes
Splenomegaly	Yes
Cytopenia (affects peripheral blood cells of lineage 2 or 3)	
HGB ≤ 9 g/dL, PLT ≤ 100 × 10^9^/L NEUT ≤ 1 × 10^9^/L	10.1 40 × 10^9^ 0.42 × 10^9^
Hypofibrinogenemia (≤1.5 g/L) or	0.74
hypertriglyceridemia (≥3.0 mmol/L or 2.65 g/L)	2.83
Hyperferritinemia (≥500 ng/mL) Increased level of soluble CD25 (soluble interleukin (IL-2) receptor (sIL-2R)) (normal range: 458–1997 pg/mL)	>33,511 12,999.1
Hemophagocytosis in bone marrow lymph nodes, spleen, or liver	Yes
Low or absent NK cell cytotoxicity (normal range > 15.11%)	12.26

## Data Availability

The original contributions presented in the study are included in the article/Appendix A; further inquiries can be directed to the corresponding author/s.

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
