# Peer review of "Hemophagocytic Lymphohistiocytosis Associated with Synergistic Defects of AP3B1 and ATM Genes: A Case Report and Literature Review"

_jcm, 2022, doi:10.3390/jcm12010095_

Round 1
Reviewer 1 Report
Yin et al. report an interesting and highly relevant case of fulminant Hemophagocytic Lymphohistiocytosis, in which the patient was succesful treated. The case is well presented. Some minor issues remain:
- The manuscript should be revised by a native speaker, e.g. „However, the increasing inci-54 dence of primary HLH in adolescents and adults [10].“
- Please specify „broad-spectrum antibi-65 otics at a local clinic“
- Please specify „high-flow oxygenation“, what was the rate of oxygen?
- Please specify „antibiotics such as line-113 zolid and carbapenem“
- What exactly is meant by „This prompted the consideration of hematologic diseases“? What exactly lead to assuming hematological diseases?
- The historical name „hemophagocytic syndrome“ should be replaced by the official nomenclature „HLH“
- The trigger search was well done.
- The authors need to discuss why a bone marrow biopsy was performed before ferritin was measured.
- „HLH is difficult to distinguish from sepsis…“ The authors should clearly state the algorithm to distinguish sepsis and HLH -> measuring ferritin and determine HLH-2004, which a safe way to diagnose HLH
- The patient met 8 out of 8 HLH-2004 criteria. It should be discussed whether the patient could have been diagnosed earlier (e.g. when still „only“ meeting 5 criteria)
- It is unclear what is meant by „HPS-II“
- The improtant value of ferritin as a screening parameter should be mentioned and discussed.
Author Response
Point 1: The manuscript should be revised by a native speaker, e.g. „However, the increasing incidence of primary HLH in adolescents and adults.“
Response 1: Thanks for the mistakes pointed out by the reviewer. Our revised content is as follows: However, the incidence of primary HLH in adolescents and adults is increasing.
Point 2: Please specify „broad-spectrum antibiotics at a local clinic“
Response 2: Thank you for your comments. The broad-spectrum antibiotics were imipenem-cilastatin sodium at a local clinic.
Point 3: Please specify „high-flow oxygenation“, what was the rate of oxygen?
Response 3: Thank you for your comments. HFNO (high flow nasal oxygenation) flow was 20 L/min, and FiO2 was 50%.
Point 4: What exactly is meant by „This prompted the consideration of hematologic diseases“? What exactly lead to assuming hematological diseases?
Response 4: Thank you for the advice. In our case, the patient presented with trilineage reduction of peripheral blood cells, persistent high fever, and splenomegaly. Laboratory tests revealed a sharp increase in ferritin, with a value of over 33,511ng/mL. This prompted the consideration of hematologic diseases,
Point 5: The historical name „hemophagocytic syndrome“ should be replaced by the official nomenclature „HLH“
Response 5: We are grateful for the suggestion. We replace “hemophagocytic syndrome” with the official name “HLH” in the revised manuscript.
Point 6: The authors need to discuss why a bone marrow biopsy was performed before ferritin was measured.
Response 6: Thank you for your advice. The patient was examined for ferritin on the day of admission, and the ferritin value was over 33,511ng/ml. The clinical manifestations of the patient were trilineage reduction of peripheral blood cells and high fever, suggesting that the patient may have hematological diseases. A bone marrow biopsy was performed to screen for hematological disorders. On the other hand, hyperferritinemia is common in sepsis, and bone marrow biopsy is used to better distinguish HLH from sepsis.
Point 7: „HLH is difficult to distinguish from sepsis…“ The authors should clearly state the algorithm to distinguish sepsis and HLH -> measuring ferritin and determine HLH-2004, which a safe way to diagnose HLH
Response 7: Thank you for your advice. HLH is difficult to distinguish from sepsis, MODS, and other cytokine storm syndromes, but the severity of the disease is often beyond the expectations of critical illness. The HLH-2004 guidelines include ferritin over 500μg/L as one of the diagnostic criteria, with a sensitivity of 84% [1]. Notably, unlike patients with sepsis, septic shock, and other conditions, patients with HLH have the highest levels of ferritin. In a retrospective study, the cutoff value of ferritin level was 9,083μg/L (sensitivity 92.5%, specificity 91.9%) to distinguish hyperferritinemia caused by HLH and sepsis or other diseases in a mixed ICU cohort [2]. In addition, bone marrow aspiration is the test most likely to lead to HLH identification [3].
Point 8: The patient met 8 out of 8 HLH-2004 criteria. It should be discussed whether the patient could have been diagnosed earlier (e.g. when still „only“ meeting 5 criteria)
Response 8: Thank you for your advice. The patient presented with the typical clinical features of HLH, including fever, hepatosplenic enlargement, and cytopenia. Laboratory results showed a dramatic increase in ferritin (> 33511ng/ml). Ferritin has significant predictive value in the diagnosis of HLH. HLH can be diagnosed early when elevated serum ferritin is associated with fever, cytopenia, and abnormal changes in the liver and spleen. At the same time, when critically ill patients have an uncontrolled inflammatory response that is difficult to explain by sepsis, HLH should be highly suspected for timely diagnosis.
Point 9: It is unclear what is meant by „HPS-II“
Response 9: Hermansky Pudlak syndrome Type 2 (HPS-II) is a rare autosomal-recessive genetic disorder caused by mutations in the AP3B1 gene [4]. It is a kind of primary HLH, which is mainly associated with immune deficiency. The clinical manifestations of HPS-II include decreased pigmentation (albinism) with visual impairment, blood platelet dysfunction with prolonged bleeding, and pulmonary fibrosis [5].
Point 10: The important value of ferritin as a screening parameter should be mentioned and discussed.
Response 10: Thank you for your suggestion. Hyperferritinemia is an essential diagnostic criterion for HLH and has been regarded as an efficient and convenient indicator for suspected HLH cases. Ferritin is an acute phase reactant that is more common in inflammation and infection. Extensive macrophages activate and release intracellular ferritin, which may be responsible for increased ferritin levels [6]. Elevated ferritin levels are seen in various conditions, such as sepsis, stroke, acute respiratory distress syndrome, renal failure, malignancy, liver disease, alcohol excess, and metabolic syndrome [7,8]. Notably, unlike patients with sepsis, septic shock, and other conditions, patients with HLH have the highest levels of ferritin [2]. The HLH-2004 guidelines include ferritin over 500μg/L as one of the diagnostic criteria, with a sensitivity of 84% [1]. The cut-off value for ferritin diagnosis varies for patients with HLH of different age groups. A ferritin over 10,000μg/L is 90% sensitive and 96% specific for diagnosing HLH in children [6]. In adults, drastically elevated ferritin levels are more meaningful for diagnosing HLH. A retrospective observational study indicated that ferritin levels above 10,000 μg/L and 100,000 μg/L predicted approximately 14% and 61% of adult patients with HLH, respectively [9].
References
- Henter, J. I.; Horne, A.; Arico, M.; Egeler, R. M.; Filipovich, A. H.; Imashuku, S.; Ladisch, S.; McClain, K.; Webb, D.; Winiarski, J.; Janka, G., HLH-2004: Diagnostic and therapeutic guidelines for hemophagocytic lymphohistiocytosis. Pediatr Blood Cancer 2007, 48 (2), 124-131.
- Lachmann, G.; Knaak, C.; Vorderwulbecke, G.; La Rosee, P.; Balzer, F.; Schenk, T.; Schuster, F. S.; Nyvlt, P.; Janka, G.; Brunkhorst, F. M.; Keh, D.; Spies, C., Hyperferritinemia in Critically Ill Patients. Crit Care Med 2020, 48 (4), 459-465.
- Raschke, R. A.; Garcia-Orr, R., Hemophagocytic lymphohistiocytosis: a potentially underrecognized association with systemic inflammatory response syndrome, severe sepsis, and septic shock in adults. Chest 2011, 140 (4), 933-938.
- Dell'Acqua, F.; Saettini, F.; Castelli, I.; Badolato, R.; Notarangelo, L. D.; Rizzari, C., Hermansky-Pudlak syndrome type II and lethal hemophagocytic lymphohistiocytosis: Case description and review of the literature. J Allergy Clin Immunol Pract 2019, 7 (7), 2476-2478 e5.
- Hasegawa, J.; Uchida, Y.; Mukai, K.; Lee, S.; Matsudaira, T.; Taguchi, T., A Role of Phosphatidylserine in the Function of Recycling Endosomes. Front Cell Dev Biol 2021, 9, 783857.
- Allen, C. E.; Yu, X.; Kozinetz, C. A.; McClain, K. L., Highly elevated ferritin levels and the diagnosis of hemophagocytic lymphohistiocytosis. Pediatr Blood Cancer 2008, 50 (6), 1227-1235.
- Koperdanova, M.; Cullis, J. O., Interpreting raised serum ferritin levels. BMJ 2015, 351, h3692.
- Kell, D. B.; Pretorius, E., Serum ferritin is an important inflammatory disease marker, as it is mainly a leakage product from damaged cells. Metallomics 2014, 6 (4), 748-773.
- Otrock, Z. K.; Hock, K. G.; Riley, S. B.; de Witte, T.; Eby, C. S.; Scott, M. G., Elevated serum ferritin is not specific for hemophagocytic lymphohistiocytosis. Ann Hematol 2017, 96 (10), 1667-1672.

Reviewer 2 Report
Major comments:
Interesting case report. HPS-II caused by AP3B1 gene mutation and ataxia telangiectasia caused by ATM gene mutation are autosomal recessive diseases. The patient was confirmed to have both AP3B1 and ATM gene heterozygous mutations, with his father having an AP3B1 gene mutation and his mother having an ATM gene mutation. The authors suspected that the synergistic defects of heterozygous mutations in AP3B1 and ATM may lead to impaired protein function, which in turn contributes to the development of HLH. This speculation partly based on reference 12 (however, a case of AT developed HLH had in fact homozygous ATM gene mutation). Although it is a critical question how heterozygous AP3B1 and ATM gene mutation is responsible for the development of clinical manifestation of HLH in this patient, it seems to be better revising the title more specific as “Hemophagocytic lymphohistiocytosis associated with synergistic defects of AP3B1 and ATM gene,” rather than current one. In Discussion, definition of HLH-associated biallelic mutations remains unclear if it limits to the same gene homozygous/compound heterozygous mutations or includes also synergistic two different heterozygous gene mutations.
Minor comments:
1. Line 54-55: However, the increasing incidence of primary HLH in adolescents and adults [10]. This part is not a sentence.
2. Reference 12; Celiksoy MH, Cubuk PO, Guner SN, Yildiran A.
3. Line 235; an AP3B1 novo mutation; de novo mutation?
4. Line252-253; Although there is no direct evidence that mutations in AP3B1 and ATM cause HLH, both are closely associated with HLH. Generally speaking, ATM seems not to be closely associated with HLH?

Author Response
Point 1: It seems to be better revising the title more specific as “Hemophagocytic lymphohistiocytosis associated with synergistic defects of AP3B1 and ATM gene,” rather than current one.
Response 1: Thank you for your suggestion. We revised the title as follows: “Hemophagocytic Lymphohistiocytosis Associated with Synergistic Defects of AP3B1 and ATM Gene: A Case Report and Literature Review”.
Point 2: Line 54-55: However, the increasing incidence of primary HLH in adolescents and adults. This part is not a sentence.
Response 2: Thanks for the mistakes pointed out by the reviewer. Our revised content is as follows: However, the incidence of primary HLH in adolescents and adults is increasing.
Point 3: Reference 12; Celiksoy MH, Cubuk PO, Guner SN, Yildiran A.
Response 3: Thanks for the mistakes pointed out by the reviewer. We have revised it in the manuscript.
Point 4: Line 235; an AP3B1 novo mutation; de novo mutation?
Response 4: Thanks for the mistakes pointed out by the reviewer. Our revised content is as follows: In this case, an AP3B1 de novo mutation did not lead to HPS-II.
Point 5: Line252-253; Although there is no direct evidence that mutations in AP3B1 and ATM cause HLH, both are closely associated with HLH. Generally speaking, ATM seems not to be closely associated with HLH?
Response 5: We agree with your valuable views. ATM gene is not closely associated with HLH. Currently, only one case of a 3-year-old male infant with an ATM gene mutation has been reported to develop HLH during follow-up. We first report and provide a clinical reference for further exploration of the mechanism. More research is needed to explore the association between ATM and HLH.
